# Mapping ActEarly: Using a child health map to evaluate a City Collaboratory programme on early promotion of good health and wellbeing

Patience Gansallo[1], Maria Bryant[2,3]*, Liina Mansukoski[2,4], Louise Padgett[2], Jessica Sheringham[5], Shahid Islam[6], Philip Garnett[4]

1 Public health and Prevention Department, Doncaster Council, Doncaster, United Kingdom,
2 Department of Health Sciences, University of York, York, United Kingdom, 3 Hull York Medical School, University of York, York, United Kingdom, 4 School for Business and Society, University of York, York, United Kingdom, 5 Primary Care and Population Health, University College London, London, United Kingdom, 6 Bradford Institute of Health Research, Bradford Royal Infirmary, Bradford, United Kingdom

* maria.bryant@york.ac.uk

## Abstract

### Background

Public health is increasingly being viewed as the result of numerous interrelated elements within intricate systems. A systems mapping approach highlights the potential direct and indirect impacts of public health programmes, the contexts within which they take place and the relations between the multiple factors at play. We report on an approach to evaluate the extent to which a city-wide programme of interventions, delivered in two locations to improve child health (ActEarly) provided activities across the child health system and addressed early-life core outcomes which were selected based on priorities identified by key stakeholders.

### Methods

Data from an 'ActEarly project log' and other information sources were used to gather a detailed picture of 68 projects that were delivered within the programme. We then used a matrix approach to map these activities against activities and outcomes from an existing child health map (CHM) of the determinants of child health inequalities and a project specific public health core outcome set (COS), developed by community and policy maker input, for systems-wide promotion of early life health and wellbeing. This was conducted alongside the creation of a new systems map and a network analysis to highlight how ActEarly operated across the children's health system (scaled in proportion to the number of projects).

**Data availability statement:** The database (matrix) supporting systems mapping, plus a detailed list of projects and interactive map are available from the OSF platform at https://osf.io/g4jf7/overview.

**Funding:** Authors MB, JS and SI received a UK Prevention Research Partnership (MR/S037527/1) award for this study/project, which is funded by the British Heart Foundation, Cancer Research UK, Chief Scientist Office of the Scottish Government Health and Social Care Directorates, Engineering and Physical Sciences Research Council, Economic and Social Research Council, Health and Social Care Research and Development Division (Welsh Government), Medical Research Council, National Institute for Health Research, Natural Environment Research Council, Public Health Agency (Northern Ireland), The Health Foundation and Wellcome.

**Competing interests:** The authors have declared that no competing interests exist.

## Results and conclusions

The map showed substantial ActEarly activity across all six CHM domains (95/139 factors) with most projects targeting the service and governance domains. There was a focus on service/governance areas of the child health system, rather than individual behaviour change, which aligned with ActEarly's objective of influencing structural barriers to health.

Projects also mapped well onto 32 of the 35 outcomes across the COS domains, with the exception of Adult Obesity, Safety at Home, and Domestic Abuse. This suggests that ActEarly aligned well with the priority outcomes from local representatives, partners and community groups.

## Introduction

A child's health is inextricably linked to their environment. This includes the services that children can access and the standard of care they receive from the health system [1]. Exposure to adverse economic, physical, cultural, educational, social, service and environmental risk factors is consistently greater in those living in areas where child poverty is high [2]. These factors can predispose children and their families to poorer mental and physical health outcomes [2]. Thus, children living in deprived areas are more susceptible to various aspects of inequalities in health. These broader determinants and disparities often intersect to reduce life opportunities [3].

Disparities in health between children from affluent and underprivileged families continue to increase in the United Kingdom [1]. This is unfortunately reflected even in child mortality rates, with mortality rates for children resident in the most deprived neighbourhoods of England (48.1 per 100,000 population) being more than double that of children resident in the least deprived neighbourhoods (18.7 per 100,000 population) [4]. Rising inequalities also exist across most health outcomes such as obesity, oral and mental health [5,6]. Public health interventions focused on a whole systems approach are advocated because child health outcomes are a product of complex and inter-related social, personal, political and economic factors [7]. Systems thinking encourages consideration of the relationships between various parts of the system (individuals, populations, or organisations) as well as the potential effects of actions in one system part on another. When considering public health issues, adopting a systems lens can help with policy and programme evaluation, as well as development of interventions that consider broader system implications [8,9].

In 2019, a UKPRP-funded (UK Preventative Research Partnership) city-wide programme of research called 'ActEarly' was launched with a focus on early life changes to improve the health and wellbeing for children living in two different areas with high rates of child poverty: Bradford, West Yorkshire and the London Borough of Tower Hamlets (LBTH) [10]. The programme was a collaboration between academics, staff and students from associated universities (University of York, Leeds, Bradford, Queen Mary University London, University College London, London School

of Hygiene and Tropical Medicine), local governments, the National Health Service (NHS), Bradford Institute for Health Research, and community and third sector organisations. The research combined interventions with citizen science and the co-production of research with local communities [11,12]. ActEarly worked in recognition that childhood and adolescence are crucial life stages that shape an individual's health and well-being throughout life [8]. It aimed to investigate health issues by examining poor and unequal health as outcomes of a multitude of interdependent elements within a connected whole from a systems perspective [13,14]. Thus, ActEarly took into consideration the key concepts of complex adaptive systems in trying to reduce health disparities through focused programmes [10,15].

ActEarly consisted of 68 independently funded projects, interventions and partnerships. These were focused on varying early life and childhood themes (e.g., healthy schools, healthy environments, healthy livelihoods) and located across the two sites (West Yorkshire and London). Individual projects were evaluated using study designs relevant to their research questions. Some consisted of large projects with sub-projects. Academics in ActEarly from different disciplines collaborated with a variety of partners at both sites, including schools, local authorities, and voluntary and community organisations to engage in research, host knowledge-sharing events and enhance routine data access. Various actors collaborated to co-design evaluations and support access to data through enabling data sharing and availability of data. An example of such partnership with ActEarly was the JU:MP (Join Us: Move: Play) project [16], embedded within the play and physical activity theme, developed to help children and families in North Bradford to move and play more. Another example was the CoPPER (Development of the Centre for Co Production and Peer Research) network [17], which was formed as an ActEarly project in Bradford with an aim to develop new ways of working to help communities, researchers and organisations work together as equal partners. CoPPER facilitates local research by exploring what communities feel are positive and negative about the areas they live in and suggesting improvements that might be made.

As part of the ActEarly systems evaluation [18], we sought to explore the extent to which ActEarly operated across the differing domains the complex system of child health and wellbeing, by mapping out the projects, interventions and partnerships within an existing child health system map by COY and colleagues; hereon called the Child Health Map (CHM) [7]. Considering ActEarly overarching goal of enacting city-wide changes across the child health system, we were therefore interested in how ActEarly operated across this system utilising the Child health map and the Core outcomes set for early life [2]. The existing CHM produced by Jessiman and colleagues [7] was designed to better understand the complexity around child health inequalities and demonstrate how individual policies or interventions may or may not effectively impact on the various dynamic issues of child health for those aged 0–25 years as indicated in the NHS [7]. The CHM was designed to better understand the complexity around child health inequalities and demonstrate how individual policies or interventions may or may not effectively impact on the various dynamic issues of child health [7]. The authors therefore sought to identify what factors drive child health and wellbeing in a local area and what factors drive child health inequalities between different groups of children and young people in a local area. ActEarly shares this systems-oriented perspective around child health inequalities, focusing on two local areas in England with high levels of child deprivation. Our research aimed to understand and highlight which of the factors that drive child health inequalities highlighted in the CHM were most addressed/targeted by ActEarly projects, thereby operationalising the CHM into practice within local contexts. CHM development (over many years) used a group concept mapping approach which enabled partners to come to an agreement on how to depict the child health system, through the creation of a shared knowledge of the issues around inequalities amongst the people working to address them. Research participants included children and young people, their carers, and professionals from local authorities, the NHS, and CCGs, as well as representatives from third-sector organizations and elected members with expertise in child health from the two study areas. The final system map showing the determinants of child health inequalities was refined through input from experts in public health, local governance, and systems science [7]. Two local authority areas were selected to create locality-specific system maps, chosen for their contrasting geography, governance structures, and population profiles to inform the development of a generic map. Site 1 was an urban city in Northern England, while Site 2 was a rural county in the Southwest [7]. The sites differed in governance

arrangements (unitary versus two-tier), levels of deprivation based on the Indices of Multiple Deprivation [19], and ethnic diversity [20]. Commonalities between the area-specific system maps, along with the removal of locality-specific factors, indicated that the CHM was applicable for use across English localities. The resulting map was "clustered" into appropriate sub-systems or "domains" using a conceptual model with interconnected domains [21] to represent the main spheres of influence on child health, and inequitable outcomes. The Child Health Systems Map has been referenced in studies that emphasise the importance of systems thinking in addressing complex public health challenges. These works share a common recognition that understanding what works, and what does not, is essential for effective policy and programme evaluation. They argue that adopting a systems perspective supports the development of interventions that account for wider system influences, acknowledging that the factors underpinning health inequalities are inherently complex [22,23]. To the best of our knowledge, this study is the first to employ the child health system map by Jessiman and colleagues [7] as an approach for identifying areas of activity of a system-wide public health collaboration (ActEarly).

As noted earlier we were also interested to learn about whether the ActEarly projects were aligned to priority areas that were determined at the start of ActEarly by system representatives, collaborating partners and members of the community. These priority areas, or 'early life outcomes' (EL COS) [2], were co-designed using a Delphi process and community consultation to prioritise and evaluate ActEarly projects. It includes six domains with 35 key early life and child public health outcomes.

In the present study, both the early life COS and the CHM [7] were considered in relation to ActEarly individual projects to explore the extent to which the programme operated across the full child health system and within priority areas. In addition to allowing us to see how ActEarly interventions interacted with children's health more broadly, the resulting 'meta system map' would also highlight gaps in activities and localities. Similar to other systems maps, it was hypothesised that this work could support policy decisions and help to identify areas of future intervention focus, in addition to evaluating the implementation of ActEarly. Specifically, we sought to answer the following questions:

1. To what extent did ActEarly deliver activities across the child health care system (CHM)?

2. How well did ActEarly activities map onto outcomes that were prioritised by key partners and members of the community (EY-COS)?

3. How well did our COS outcomes align to those within the CHM?

## Materials and methods

### Study design

This paper describes the mapping of ActEarly activities, interventions and partnerships (hereon combined and termed as 'projects') to develop a higher-level view of where ActEarly interacted within the children's health system (CHM). A systems approach was applied to enable us to identify how various factors/outcomes were interconnected, explore their potential influence on child health and understand more broadly the extent to which inequalities could have been addressed through the projects.

### Project sites

The ActEarly projects were conducted across two study sites, with high rates of child poverty in England: Bradford (West Yorkshire), and the London Borough of Tower Hamlets (LBTH) when compared to the national average. [10]. Bradford is a post-industrial city in the North of England and ethnically diverse, with a large South Asian community. Bradford has a population of 560,200, where 25.5% of people identify as Pakistani; the second largest percentage nationally [24]. Child poverty is high, with 33.2% of children aged under 16 living within absolute low-income families. Around two thirds of the wards in Bradford had a higher proportion of relative child poverty than the national average of 19.9% [25].

LBTH has a population of 310,300 [26]. The proportion of children living in absolute low-income families was recorded at 20.7% in 2023 [27]. LBTH is ethnically diverse, having the largest Bangladeshi population in the country (107,333 residents, 34.6% of the population). Children and older people are far more likely to be living in poverty in Tower Hamlets than those living elsewhere in the country. About 47% of children live in poverty after housing costs in LBTH, substantially above the UK average of about 31% [28]. Across 20 local authority districts across England, 50% or more children and older people are reported as living in income deprived households. Tower Hamlets has the highest proportion at 71.3% for children and (61.1%) for adults [29].

## Projects

ActEarly was delivered between 2019–2025; however, it has 'legacy projects' that will continue beyond this timeline. During this main delivery timeline, it consisted of 68 projects, interventions and/or partnerships that targeted multiple, complex inequality areas in the child health system and were spanned across five main themes (Healthy Places; Healthy Learning; Food and Healthy Weight; Play and Physical Activity; Healthy Livelihoods) and two methodological, cross-cutting themes (Co-Production; and Evaluation). Some examples of projects within the main themes included Born in Bradford (BiB) Breathes, a clean air intervention project [30], JU;MP, a physical activity project [16] and Healthier Wealthier Families in East London, providing combined health and financial advice to communities. Across the 68 projects, different study designs were applied by individual project teams, including quasi experimental evaluations, citizen science, Delphi, randomised control trials, longitudinal observational research, participatory research, systematic reviews, narrative reviews, data science, systems mapping, qualitative studies, and mixed methods. The cross-cutting evaluation theme of ActEarly, as noted in the protocol [18] focused on evaluating the programme as a whole, including exploring the potential impact of the programme on early life health, identifying the extent to which it fostered new collaborations and networks and built capacity, and the degree to which it influenced policy and practice [18].

## Information sources

We used data from an 'ActEarly project log', which gathered details such as project title, timeline, theme, project type, study design and methods, details of participants and outcomes (where relevant) to obtain key project information needed to map each project against an existing child health system map (see S1 Table). Other information sources included the ActEarly website, individual project websites, activity posters, project outputs (for example: protocols, reviews, preliminary findings), ActEarly publications, monthly news and events, and policy briefs. L.N undertook the capturing of the data for each project and regularly updated the project log, which was also supported by P.G. Once a project had secured funding and approved proposal, all the necessary information was added to the project log. Any changes made to a project afterwards, were updated in the project log. There was an exception of one project which was excluded due to the funding that was not confirmed. Data for each project were continually captured and reviewed and sense checked monthly from the start of the ActEarly project (01.10.2019) to its completion (31.09.2024) to ensure relevant details and information retrieved from these sources was also sense checked with researchers who worked on the projects for further clarity where needed. Information was gathered through one-on-one meetings with research fellows working within each ActEarly theme, and through two longer meetings with senior academics and project leaders in their executive meetings. Each theme lead was asked to review the project list relevant to their theme, supplement the information given by the research fellows, and address any outstanding queries.

## Child health systems map

The existing Jessiman child health system map (CHM) that was used to map ActEarly activities against in this study includes 139 factors influencing inequalities in health outcomes for children aged 0–25 years [7]. Similar to CHM, ActEarly projects were focused on early years, school-aged children, and young adults. [10]. Jessiman and colleagues categorised

 

the CHM into six system domains, representing the varying spheres of influence on child health which are critical in reducing inequalities and improving child health and wellbeing.

**Domains within the child health systems map**

1. Service

2. Governance

3. Economic

4. Social

5. Physical

6. Personal domain [7] (Table 1)

There are multiple factors within each of the domains of the child health systems map as shown in Table 1.

**Core outcome set**

In order to explore the degree to which ActEarly also operated in areas that were prioritised by local representatives and communities, we also mapped projects on to an ActEarly specific Core Outcome Set (COS), which includes outcomes across the six domains of early life [2].

Table 1. Child health systems map (CHM) and ActEarly COS domains, and the criteria used to map ActEarly projects on to CHM and COS.

| Child Health Map (CHM) Domain | Number of factors within each domain | CHM criteria (projects were mapped to the factors in each domain based on the details provided below) |
|---|---|---|
| Service | 46 | Associated with availability and delivery of services provided by public, private or other sectors. |
| Governance | 15 | Includes conditions associated with the effective development and implementation of policy to support child health and reduce inequalities |
| Personal | 26 | Related to children and young peoples' behaviours and outcomes |
| Economic | 12 | Economic resources that are available to children and young people, at both the household and local area-level. |
| Social | 24 | Highlighting the people around children and young people who influence their health behaviours and outcomes, including parents and families, peer groups, and local communities. Additionally, the influences they encounter online and social media. |
| Physical | 16 | Related to the physical environment in which children and young people live |
| **ActEarly COS domain** | | **ActEarly COS Criteria (projects were mapped to each domain based on these outcomes)** |
| Development and education | 6 | Access to education, speech, language and communication, emotional and social development, children get the best start in life, educational attainment, access to books. |
| Physical health and health behaviours | 6 | Child physical activity, Child sedentary behaviour, Healthy eating, Child weight, Childhood obesity, adult obesity |
| Mental health | 5 | Child happiness, child mental health (incl. children's stress and anxiety), child mental well-being, parental mental health, parental mental well-being |
| Social environment | 4 | Family and social relationships, safety at home, domestic abuse, child social relationships and bullying |
| Physical environment | 7 | Use, quality, and satisfaction with open space, parks and green spaces, access to high quality health services, air pollution, food availability, quality of local environment, traffic. |
| Poverty and inequality | 7 | Housing (incl. homelessness; house crowding; availability of affordable housing), access to opportunity, basic care needs met, employment, financial stability, inequalities, poverty |

### Domains within the core outcome set

1. Development and education

2. Physical health and health behaviours

3. Mental health,

4. Social environment

5. Physical environment

6. Poverty and inequality [2] (Table 1).

There are multiple outcomes within each of the domains of the core outcome set as shown in Table 1

### Mapping of projects

We mapped the information retrieved from each ActEarly project to domains within the CHM and the ActEarly COS (Table 1). The mapping of the projects and further analysis was undertaken by P.G, while P.G conducted the network analysis. M.B and L.N. L.P, J.S, S.I contributed to reviewing of the analysis. A deduction framework approach was applied, and a matrix was created to code all projects to factors in the CHM and the COS outcomes, with projects listed on the vertical Y-axis and the factors/outcomes on the horizontal X-axis (See S2 File CHM matrix and S1 File COS matrix) using criteria agreed by Jessiman et al, during the creation of the child health systems map (Table 1). The matrix was developed using Microsoft Excel with two specific headings (one for projects and the other factors/outcomes), These were reviewed and sense-checked by the research team to ensure consistency and accuracy. A score of 1 was assigned to any project that was linked to a factor/outcome and 0 indicated none. We then summarised and ranked the number of ActEarly projects linked to each factor/outcome within the domain. This enabled us to identify areas in the CHM which were likely to be impacted by the ActEarly programme (i.e., the 'dose' of projects within each domain).

### Network analysis

Analysis calculated the degree centrality of ActEarly factors (activities) in the CHM (the number of connections between that factor and any other factors in the system) as this may provide an indication of the significance of that part of the system to the system as a whole, as well as the number of ActEarly projects operating within each CHM factor. We also calculated the number of ActEarly projects that mapped on to core outcomes within the Early Life COS. To generate the map, we reproduced the original CHM published by Jessiman et al [7] in iGraph in R (using a script to convert the original vue-files into gml-filea (https://github.com/prgarnett/vueparser)). Fig 1 was then created using a Fruchterman-Reingold layout algorithm [31]. This layout algorithm positions nodes based on the number of relationships they have with other nodes, therefore denser (more highly connected) parts of the network tend to group together. We then scaled the size of the nodes in the network in proportion to the number of ActEarly projects that operated in that part of the children's health system (using the log of the count to avoid very large nodes). This provided a relatively straightforward way of visually seeing areas of high vs low ActEarly activity in the system.

## Results

Results of the mapping of projects on both CHM and COS domains can be found in Tables 2 and 3. The CHM has a total of 139 factors influencing child health inequality and our findings showed that projects were mapped onto 95 of these (and therefore not aimed at 44 factors across all domains). The CHM factors which had activity in at least one ActEarly project are shown in Table 2 according to the CHM domain.

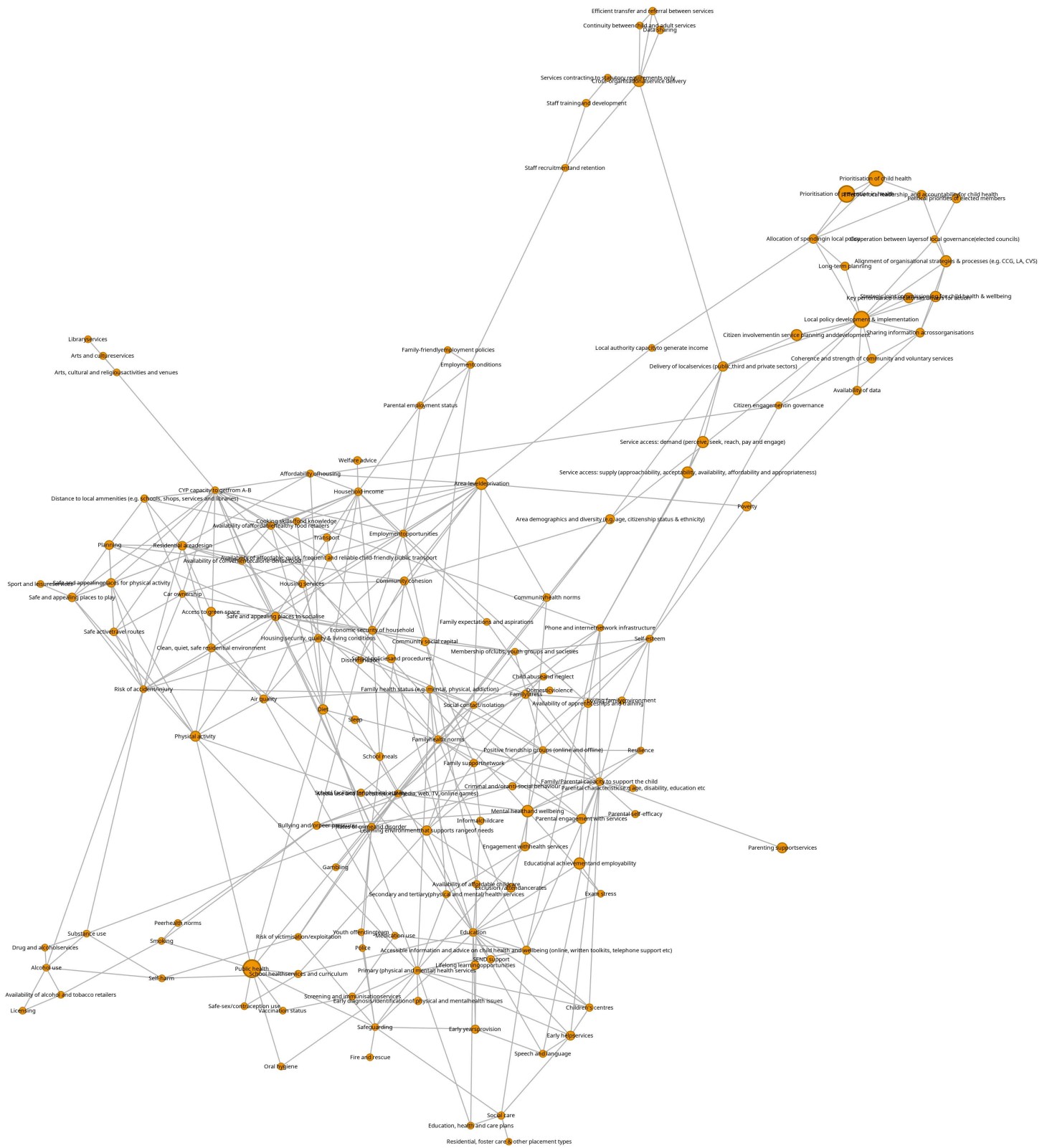

**Fig 1. Systems map highlighting ActEarly activity within the Child Health Map.**

**Table 2. Factors targeted by projects across each domain.**

| CHM Domain/ number of factors | Number (%) of CHM factors targeted by at least one AE project | Specific factors within the CHM that were a focus of at least one ActEarly project |
|---|---|---|
| Service/ 46 | 32 (70%) | School facilities for physical activity, SEND Support, Lifelong learning opportunities, School meals, Education, School policies and procedures, School health services and curriculum, Learning environment that supports range of needs, Early years provision, Citizen involvement in service planning and development, Availability of affordable childcare, Service access:demand, Service access:supply, Staff training and development, Staff recruitment and retention, Data sharing, Cross-organisational service delivery, Efficient transfer and referral between services, Delivery of local services, Parental support services, Planning, Transport, Welfare advice, Primary (physical and mental health services), Secondary and tertiary (physical and mental health services), Public health, Screening and immunization services, Early diagnosis of physical/mental health issues, Accessible information and advice on child health, Social care, Early help services, Housing services. |
| Governance/ 15 | 14 (93%) | Alignment of organisational strategies and processes, Strategic joint commissioning of child health and wellbeing, Sharing information across organisations, Availability of data, Local policy development and implementation, Coherence and strength of community and voluntary services, Cooperation between layers of local governance, Political priorities of elected members, Key performance indicators as drivers for child health, Long term, Planning, Prioritisation of prevention in health, Prioritisation of child health, Allocation of spending in local policy, Effective local leadership and accountability for child health. |
| Personal/ 26 | 13 (50%) | Criminal and/or antisocial behaviour, Self-esteem, Cooking skills/knowledge, Diet, Mental health and wellbeing, Physical activity, Sleep, Engagement with health services, Exam related stress, Exclusion/attendance rates (school), Educational achievement/employability, Speech and language skill, Risk of accident/injury |
| Economic/ 12 | 8 (67%) | Household income, Economic security of household, Affordability of housing, Area-level deprivation, Poverty, Employment opportunities, Employment conditions, Availability of apprenticeships/ training. |
| Social/ 24 | 16 (67%) | Family/Parental capacity to support the child, Family Stress, Parental engagement with services, Family health norms, Informal childcare, Family support network, Family health status, Parental self-efficacy, Community social capital, Community cohesion, Area demographics, Community health norms, Rates of crime and disorder, Bullying and/or peer pressure, Media use and influence, Loving family environment. |
| Physical/ 16 | 12 (75%) | Safe and appealing places to socialise, Safe and appealing places for physical activity, Safe and appealing places to play, Access to green spaces, Arts, cultural and religious activities and venues, Safe active travel routes, Distance to local amenities, Residential area design, Housing security, quality and living conditions, Clean, quiet, safe residential environment, Air quality, Availability of affordable healthy food retailers. |

Table 2 shows that all domains of the CHM were represented by ActEarly projects. The governance domain, with 15 factors, revealed a wide coverage of projects across 14 factors. The personal domain showed the least coverage by ActEarly activities, with only 13 records of ActEarly projects out of a potential 26. In terms of activity within areas that were prioritised by local representatives, our findings revealed that projects worked across all six domains of the COS and in 32 of the 35 specific outcomes (Table 3). Domains with the most ActEarly activities were, mental health, physical environment and poverty/inequality domains, in which all individual outcomes were covered by ActEarly activities.

Fig 1 shows the results of the network analysis process, with this version showing the nodes representing parts of the children's health system scaled in proportion to the log of the count of ActEarly projects that operate in each domain area. This shows two zones of ActEarly activity. The first is represented at the bottom right of the diagram, where there were a number of parts of the children's health system that group together that many ActEarly projects map to. This area has a concentration of system parts such as strategy, planning, leadership, and policy. The second, more central area, corresponds to a more diverse number of parts of the system.

Network data are also presented in Table 4, which presents activities in descending order by the number of projects in that part of the system. This highlights that activities related to public health factor within the child health map were most common across ActEarly. Connectivity between parts of the system suggests that those factors are connected and/or similar in some way, which may indicate that there is more potential for systems change in that general area of the system (though this would need to be confirmed with impact data).

**Table 3. ActEarly outcomes targeted by projects across each Core Outcome Set (COS) domain.**

| COS Domain/ number of outcomes | Number (%) of outcomes targeted by at least one AE project | Specific Outcomes within the Early Life COS that were a focus of at least one ActEarly project |
|---|---|---|
| Development and education/ 6 | 6 (100%) | Access to Education, Access to books, Speech, language and communication, Emotional and social development, Children get the best start in life, Educational attainment. |
| Physical health and health behaviours/ 6 | 5 (83%) | Physical-activity, Child sedentary behaviour, Healthy eating, Child weight, Childhood obesity. |
| Mental health/ 5 | 5 (100%) | Child-happiness, Child-mental health, Child-mental-wellbeing, Parental-mental-health, Parental-mental wellbeing. |
| Physical environment/ 7 | 7 (100%) | Use, quality, and satisfaction with open space, Parks and green spaces, Access to high quality health services, Air pollution, Food availability, Quality of local environment, Traffic (including traffic levels outside schools, parking). |
| Social environment/ 4 | 2 (50%) | Family and social relationships, Child social relationships or bullying. |
| Poverty and inequality/ 7 | 7 (100%) | Housing, Access to opportunity, Basic care needs met, Employment, Financial stability, Inequality, Poverty. |

Table 4 also shows the nodes of the CHM, their degree (the number of links between that node and other parts of the system), the count of the number of ActEarly projects that map to that node in system, and the number of outcomes in the COS that map to that part of the system via ActEarly projects. ActEarly demonstrated a reasonable breadth of connected activity across the CHM. Data in Table 4 shows that more than 40 ActEarly projects map to parts of the child health system including Public Health, Prioritisation of Prevention Health, Local policy development and implementation, and Prioritisation of child health. This fits with the focus of ActEarly on prevention research in public health.

There were areas of the health system with 20 or more projects aligned to them that are related to strategy, service provision, and service coordination. Ten or more projects mapped to other key focus areas for ActEarly, such as education, green spaces, child physical health, and diet. Areas of the CHM with ActEarly projects with the greatest system links (net degree) were in factors associated with local policy development and implementation, and area level deprivation. Child health system factors that were least connected by ActEarly included projects focused on areas such as parenting support services, alcohol, gambling, library services and medication.

For the core outcome set data, we found a generally good coverage of projects to core outcomes. Similar to the CHM system mapping, more core outcomes were mapped to the areas of public health, planning, and strategy.

## Discussion

This paper presents the results of a novel mapping approach to highlight areas of activity in a large system-wide public health collaboration, ActEarly. We showed that, by mapping the projects included in the programme onto an existing map of the child health system (CHM) [7], we could gain a better understanding of where within the wider system, ActEarly operated. This approach revealed both areas of high activity (e.g., policy and implementation) as well as areas of less activity (e.g., parenting support services, alcohol and gambling).

Overall, we can see that, when taking the wider aims of ActEarly into account [10], projects within ActEarly mapped well to relevant areas of the child health system. If we look at areas of the system where there were no ActEarly projects directly targeting a factor/s, it tended to be (with a couple of exceptions) where the focus was more on adults or older children, or those that link to behaviour change. For example, parental issues such as employment and access to cars, issues that would be more associated with either older children (young adults), or parents such as smoking and alcohol use in the household. In addition, some of these factors reside within the personal domain (for example social isolation, smoking) of the CHM and were included to illustrate how they have been linked to influencing other factors [7]. Due to this

**Table 4. ActEarly projects and core outcomes linked to the CHM.**

| Factors in the CHM System | Net Degree[1] | Number of ActEarly Projects | COS Mapped to System |
|---|---|---|---|
| Public health | 9 | 57 | 29 |
| Prioritisation of prevention in health | 2 | 50 | 28 |
| Local policy development & implementation | 13 | 48 | 28 |
| Prioritisation of child health | 3 | 43 | 28 |
| Area-level deprivation | 14 | 25 | 24 |
| Mental health and wellbeing | 7 | 24 | 25 |
| Educational achievement and employability | 3 | 23 | 23 |
| Service access: demand (perceive, seek, reach, pay and engage) | 3 | 23 | 21 |
| Service access: supply (approachability, acceptability, availability, affordability and appropriateness) | 3 | 23 | 21 |
| Alignment of organisational strategies & processes (e.g., CCG, LA, CVS) | 5 | 22 | 23 |
| Citizen involvement in service planning and development | 2 | 22 | 26 |
| Cross-organisational service delivery | 6 | 22 | 24 |
| Strategic joint commissioning for child health & wellbeing | 4 | 19 | 25 |
| Parenting support services | 1 | 17 | 20 |
| Physical activity | 9 | 17 | 21 |
| Key performance indicators as drivers for action | 2 | 15 | 23 |
| Learning environment that supports range of needs | 8 | 15 | 18 |
| Delivery of local services (public, third and private sectors) | 6 | 14 | 16 |
| Diet | 9 | 14 | 15 |
| Parental engagement with services | 5 | 14 | 20 |
| Area demographics and diversity (e.g., age, citizenship status & ethnicity) | 5 | 13 | 20 |
| Poverty | 3 | 13 | 17 |
| Political priorities of elected members | 2 | 12 | 19 |
| Access to green space | 3 | 11 | 18 |
| Allocation of spending in local policy | 6 | 11 | 15 |
| Early help services | 6 | 11 | 15 |
| Planning | 7 | 11 | 20 |
| Coherence and strength of community and voluntary services | 3 | 10 | 17 |
| Engagement with health services | 5 | 10 | 16 |
| Exclusion/ attendance rates | 3 | 10 | 12 |
| Residential area design | 9 | 10 | 18 |
| Sharing information across organisations | 5 | 10 | 17 |
| Availability of data | 3 | 9 | 16 |
| Effective local leadership and accountability for child health | 4 | 9 | 12 |
| Lifelong learning opportunities | 1 | 9 | 15 |
| Long-term planning | 2 | 9 | 15 |
| Safe and appealing places for physical activity | 6 | 9 | 18 |
| School policies and procedures | 4 | 9 | 16 |
| Accessible information and advice on child health and wellbeing (online, written toolkits, telephone support etc) | 8 | 8 | 12 |
| Early years provision | 4 | 8 | 18 |
| Safe and appealing places to play | 6 | 8 | 18 |
| Safe and appealing places to socialise | 11 | 8 | 18 |
| Availability of apprenticeships and training | 3 | 7 | 10 |

*(Continued)*

**Table 4.** (Continued)

| Factors in the CHM System | Net Degree[1] | Number of ActEarly Projects | COS Mapped to System |
|---|---|---|---|
| Community cohesion | 7 | 7 | 11 |
| Community social capital | 4 | 7 | 11 |
| Family stress | 8 | 7 | 14 |
| Air quality | 7 | 6 | 11 |
| Clean, quiet, safe residential environment | 8 | 6 | 13 |
| Economic security of household | 5 | 6 | 10 |
| Education | 18 | 6 | 9 |
| Community health norms | 4 | 5 | 9 |
| Data sharing | 3 | 5 | 6 |
| Household income | 14 | 5 | 10 |
| Housing security, quality & living conditions | 8 | 5 | 15 |
| School meals | 3 | 5 | 7 |
| Sleep | 2 | 5 | 17 |
| Welfare advice | 1 | 5 | 10 |
| Availability of affordable childcare | 4 | 4 | 6 |
| Bullying and/or peer pressure | 8 | 4 | 12 |
| Children's centres | 5 | 4 | 6 |
| Family health norms | 10 | 4 | 9 |
| Family support network | 5 | 4 | 16 |
| Family/Parental capacity to support the child | 19 | 4 | 14 |
| Housing services | 3 | 4 | 9 |
| Safe active travel routes | 5 | 4 | 11 |
| SEND support | 1 | 4 | 12 |
| Availability of affordable healthy food retailers | 8 | 3 | 7 |
| Criminal and/or anti-social behaviour | 3 | 3 | 17 |
| Distance to local ammenities (e.g., schools, shops, services and libraries) | 5 | 3 | 13 |
| Early diagnosis/identification of physical and mental health issues | 1 | 3 | 7 |
| Efficient transfer and referral between services | 3 | 3 | 11 |
| Informal childcare | 2 | 3 | 6 |
| Primary (physical and mental) health services | 13 | 3 | 11 |
| Rates of crime and disorder | 12 | 3 | 17 |
| School health services and curriculum | 5 | 3 | 8 |
| Self-esteem | 8 | 3 | 11 |
| Social care | 5 | 3 | 11 |
| Staff training and development | 2 | 3 | 10 |
| Transport | 1 | 3 | 7 |
| Affordability of housing | 4 | 2 | 6 |
| Cooking skills/ food knowledge | 3 | 2 | 10 |
| Employment conditions | 5 | 2 | 7 |
| Employment opportunities | 9 | 2 | 7 |
| Family health status (e.g., mental, physical, addiction) | 9 | 2 | 9 |
| Loving family environment | 3 | 2 | 10 |
| Parental self-efficacy | 2 | 2 | 4 |

*(Continued)*

**Table 4.** (Continued)

| Factors in the CHM System | Net Degree[1] | Number of ActEarly Projects | COS Mapped to System |
|---|---|---|---|
| School facilities for physical activity | 3 | 2 | 9 |
| Secondary and tertiary (physical and mental) health services | 3 | 2 | 8 |
| Speech and language | 4 | 2 | 7 |
| Arts and culture services | 1 | 1 | 0 |
| Arts, cultural and religious activities and venues | 3 | 1 | 7 |
| Cooperation between layers of local governance (elected councils) | 3 | 1 | 0 |
| CYP capacity to get from A-B | 16 | 1 | 4 |
| Exam stress | 3 | 1 | 6 |
| Media use and influence (social media, web, TV, online games) | 9 | 1 | 2 |
| Risk of accident/ injury | 11 | 1 | 6 |
| Screening and immunisation services | 3 | 1 | 1 |
| Services contracting to statutory requirements only | 2 | 1 | 0 |
| Staff recruitment and retention | 3 | 1 | 1 |
| Alcohol use | 6 | 0 | 0 |
| Availability of affordable, quick, frequent and reliable child-friendly public transport | 5 | 0 | 0 |
| Availability of alcohol and tobacco retailers | 3 | 0 | 0 |
| Availability of convenience calorie-dense food | 7 | 0 | 0 |
| Car ownership | 7 | 0 | 0 |
| Child abuse and neglect | 1 | 0 | 0 |
| Citizen engagement in governance | 4 | 0 | 0 |
| Continuity between child and adult services | 3 | 0 | 0 |
| Discrimination | 3 | 0 | 0 |
| Domestic violence | 1 | 0 | 0 |
| Drug and alcohol services | 2 | 0 | 0 |
| Education, health and care plans | 3 | 0 | 0 |
| Family expectations and aspirations | 2 | 0 | 0 |
| Family-friendly employment policies | 2 | 0 | 0 |
| Fire and rescue | 1 | 0 | 0 |
| Gambling | 1 | 0 | 0 |
| Library services | 1 | 0 | 0 |
| Licensing | 2 | 0 | 0 |
| Local authority capacity to generate income | 2 | 0 | 0 |
| Medication use | 3 | 0 | 0 |
| Membership of clubs, youth groups and societies | 5 | 0 | 0 |
| Oral hygiene | 2 | 0 | 0 |
| Parental characteristics, e.g., age, disability, education etc | 1 | 0 | 0 |
| Parental employment status | 4 | 0 | 0 |
| Peer health norms | 2 | 0 | 0 |
| Phone and internet network infrastructure | 8 | 0 | 0 |
| Police | 2 | 0 | 0 |
| Positive friendship groups (online and offline) | 7 | 0 | 0 |
| Residential, foster care & other placement types | 1 | 0 | 0 |
| Resilience | 4 | 0 | 0 |

*(Continued)*

**Table 4.** (Continued)

| Factors in the CHM System | Net Degree[1] | Number of ActEarly Projects | COS Mapped to System |
|---|---|---|---|
| Risk of victimisation/ exploitation | 4 | 0 | 0 |
| Safe-sex/ contraception use | 2 | 0 | 0 |
| Safeguarding | 9 | 0 | 0 |
| Self-harm | 4 | 0 | 0 |
| Smoking | 4 | 0 | 0 |
| Social contact/ isolation | 11 | 0 | 0 |
| Sport and leisure services | 2 | 0 | 0 |
| Substance use | 5 | 0 | 0 |
| Vaccination status | 2 | 0 | 0 |
| Youth offending team | 2 | 0 | 0 |

[1]The number of links between that node and other parts of the system.

interlink between factors further research is needed to understand how ActEarly project mays have impacted on these other areas of the CHM".

The overarching aim of ActEarly was based on system level change, rather than individual behaviour change interventions, so this is perhaps expected (though also confirms that the programme operated in areas that it intended to operate in). Our study further revealed some examples that show the multi-level and interdependent nature of drivers of child health, as our map highlights interlinks between various factors and projects. For example, projects which had the potential to impact education, a factor in the service domain, were also linked with exclusion/attendance at schools and educational achievement/employability. This understanding has been reported by studies that explored systems mapping in evaluating specific health outcomes such as physical activity, healthy eating, cardiovascular fitness and obesity [32–34].

There were significant interlinks between ActEarly projects and factors in the CHM, with some level of overlap across the governance, physical, social and personal domains. Some examples of projects include APPG Child of the North [35], ActEarly Co-production activities [36] and JU;MP [16], thus, one project had the capacity to impact various domains. Further, Mapping ActEarly projects highlighted that working across diverse actors and systems could potentially influence development of multi-level actions. For example, cross-site (Bradford and London) projects such as "Unlocking data to improve public health policy" [37,38] had the potential to support the development of a whole system data linkage and impact factors such as access to data/availability of data.

Considering the ActEarly map, we do not classify all projects addressing social or economic aspects of child health inequalities as upstream in nature. While some aimed to tackle structural determinants such as directly influencing income or housing, few addressed this through more immediate practical measures, for example, providing welfare advice. This illustrates the complexity of system-level change, where in some situations, a project may shift toward supporting individual behaviour.

In addition to demonstrated good coverage of ActEarly across the CHM system, the projects were fairly well represented across all the domains of the early life COS [2], which were prioritised by stakeholders, including members of the public (32 of 35 outcomes). Some outcomes that were prioritised that demonstrated a wide coverage of projects included inequality, basic care needs met, poverty and access to opportunity. This highlights ActEarly's aim of working towards reducing child poverty and inequality. Outcomes such as physical activity, educational attainment, mental health/wellbeing and quality of the local environment also revealed significant ActEarly activities. Although there was an uneven distribution of the projects across the outcomes, each of the six domains of the COS showed some recorded activities. Our findings did show however, show that minimal projects targeted early child development outcomes, such as speech, language, and communication.

Most outcomes in the COS [2] overlapped with various factors in the CHM [7]. One of these was the physical environment, which was identified and captured by ActEarly projects across the COS and CHM. Examples of these were, safe places to play, access to green spaces, traffic, and air-pollution. This likely highlights ActEarly's focus on prevention research. Improving economic resources, reducing poverty and inequality were also significant areas captured by ActEarly projects across the COS and CHM. Some of these projects varied across housing, access to opportunity, welfare benefits advice, financial stability, and inequalities in early years provision. Other areas of overlap, which are key to the health and wellbeing of children and highlighted in both the COS and CHM included healthy eating/diet, food availability, educational attainment, physical activity, mental health and wellbeing. Most of the factors in the governance and service domain of the CHM associated with the effective development and implementation of policy to support child health and availability and delivery of services provided by public, private or other sectors were not prioritised by partners/stakeholders in the COS [2].

The CHM categorised mental health as a single factor within the personal domain, while the COS highlighted this as a domain with five different outcomes considering it in relation to both children and parents. Furthermore, the COS only captured four outcomes around the social environment; however, there was more evident coverage of this domain in the CHM particularly community related outcomes such as community social capital, community cohesion, all of which are reflections of the relevance of the community in child health. The COS prioritised more early years developmental outcomes when compared to the CHM. Some of these include children get better start in life, speech, language and communication and other child specific outcomes such as child weight, child obesity and child happiness. The inclusion of these outcomes in the COS but not the CHM is likely to reflect the initial use of key indicator public health frameworks that were employed in the COS development, in addition to the consultations and lived experiences of the communities.

The governance domain of the child health system captured some ActEarly activities which were significant in ensuring sustainability of the programme interventions. Examples of this include local policy development and implementation and alignment of organisational strategies and processes. This finding varied with that reported in a recent study which aimed to produce a mega-map to identify, map and provide an overview of the existing evidence synthesis on the interventions aimed at improving child well-being in low- and middle-income countries (LMICs) [39]. In their research, Saran et al revealed that most of the interventions focused on early childhood development with governance shown as the least studied area.

Jessiman et al. [7] anticipated that many of the factors and domains that were represented may not be exclusive to England alone, although this has yet to be tested or researched. The map can also be used to identify factors that may be influenced by local public health interventions and inform evaluation design" [7]. Within the context of a whole-systems approach to addressing child health inequalities, the CHM emerges as a relatively new framework designed to support strategies for reducing inequalities. In addition, we used the EY-COS [2], which facilitates the evaluation of individual interventions within system-change approaches. The EY-COS outcomes reflect priority areas identified at the start of ActEarly by system representatives, collaborating partners, and community members [2]. The EY-COS, applied in this study, enabled us to address aspects of the CHM domain that were not captured in the map. Examples include speech, language and communication, children get best start in life, child sedentary behaviour".

One of the strengths of the CHM lies in its value in demonstrating to stakeholders the need for policy approaches that address the broader systemic determinants of child health inequalities, extending beyond traditional public health interventions. In addition, the CHM also serves as a systems-thinking tool to assist Integrated Care Systems in recognising a broad spectrum of factors that contribute to child health inequalities [7]. It was however noted that the CHM is complex and potentially overwhelming, with participants suggesting the development of an interactive version to support easier interpretation and adaptation for use in local areas [7]. This limitation was considered during our study; therefore, we engaged in a detailed analytical review of the existing CHM within each specific domain, examining all factors individually before adapting them for the revised ActEarly map. It was also noted that the CHM may also need more adequate

information to understand the direction of influence between factors (as most factors are interlinked), hence during our analysis, we also considered the factors individually.

Going forward, our map (and its development process) could be used to support new insights and useful learnings for future planning and delivery of interventions and policies that are focused on child health. For example, it could help to identify elements and relationships of the system that need to be targeted (and whether the areas of highest activity are the ones where there are measurable changes in outcomes), in addition to gaps in planned intervention strategies. Mapping initiatives such as the one showcased here can also be used for creation of holistic visuals depicting the various drivers of a public health area of interest [40]. This paper provides a perspective on the use of systems mapping for the evaluation of public health programmes. It can often be challenging to see the bigger picture of a larger project, particularly while that project is still ongoing. However, this can be essential information as it can give an indication of areas the project is having an impact on the target system of interest, but perhaps as importantly it can provide insight into where the project is not present.

The use of systems mapping in investigating large projects like ActEarly also has the potential to facilitate development of shared goals and understanding, highlighting areas of recommendations for local context, including areas that may have received a lower level of investment in relation to the importance placed on them by partners and members of the community [32]. As suggested by Bronfenbrenner and Ceci [41], our results also show that systems maps can extend our thinking and indicate how multiple influences on an individual's health can be addressed. This can help us to also appreciate the interactions between the different levels of interventions, as evident in ActEarly projects. For example, interventions focused on encouraging increased physical activity could also support the improvement of mental health.

A key strength of this study is that the review of ActEarly projects and the revised map generated has taken into consideration varying outcomes in child health, in addition to aspects of child health where inequality may arise. We believe the approach described in this paper is novel in scope and could support complex evaluation of large-scale, system-level public health programmes. The systems map provides an informative way to illustrate multiple components in which ActEarly delivered projects to address child health and could be a potential guide in areas of sustainable policy. However, some level of subjectivity was applied during the process due to the varying nature of projects and information provided. Our evaluation and the information available to us does not allow us to directly investigate the accuracy of our findings but suggests areas of the wider complex system with recorded ActEarly projects and where we are most likely to observe a change. Though the domains of the child health system are interlinked, for the purpose of this study the factors in the child health system were considered individually.

In our research, degree centrality was used to indicate how interconnected ActEarly activities were within the CHM. Thus, we examined the distribution of ActEarly projects across CHM which allowed us to consider not only connectivity but also practical engagement (identifying which areas of the system were actively targeted by projects). Our research enabled us to highlight potential leverage points for system-level change, while acknowledging that significance in terms of health impact requires more evaluation. The systems map used in the study reflects a specific context, and participant set, and therefore the areas in our revised map with most ActEarly projects may not universally provide an indication of the most significant factors influencing child health inequality.

Care should also be taken to not over interpret the network, as there are no data that might indicate the strength of the impact on that area of the child health system by any individual project, nor are we able to say if there is any cumulative impact of projects either on one part of the system (any one node) or regions of the network as linked by relationships between the parts. It is merely indicative of ActEarly activity in that area. In addition, many projects operated across multiple domains or have an indirect influence, which again highlights the complexity of interpreting connectivity and project activity as measures of impact. Our aim is to provide an overview of ActEarly reach and potential leverage points, while acknowledging that causal significance and system-wide impact require further evaluation. Therefore, the map alone is not sufficient to understand how and to what extent those projects are impacting the child health system. However, it is a useful heuristic and can play a part in developing more detailed knowledge.

## Conclusion

This study used a novel approach to explore the degree to which a complex, city-wide multilevel programme was able to intervene across the child health system, and provided clarity about where activity was focused, relative to local and national priorities. Importantly, we found that ActEarly mapped well to areas of the child health system that were aligned to the objective of the programme to target upstream determinants of child health.

## Supporting information

**S1 Table. Project log: A summary of all 68 projects delivered within ActEarly.**
(DOCX)

**S1 File. Early Life Core Outcome Set matrix.**
(XLSX)

**S2 File. Child Health Map matrix.**
(XLSX)

## Author contributions

**Conceptualization:** Maria Bryant, Liina Mansukoski, Jessica Sheringham, Philip Garnett.

**Data curation:** Patience Gansallo, Liina Mansukoski, Louise Padgett, Philip Garnett.

**Formal analysis:** Patience Gansallo, Philip Garnett.

**Funding acquisition:** Maria Bryant, Jessica Sheringham.

**Investigation:** Patience Gansallo, Liina Mansukoski, Shahid Islam.

**Methodology:** Patience Gansallo, Maria Bryant, Liina Mansukoski.

**Project administration:** Maria Bryant.

**Software:** Philip Garnett.

**Supervision:** Maria Bryant.

**Validation:** Louise Padgett.

**Writing – original draft:** Patience Gansallo, Maria Bryant.

**Writing – review & editing:** Patience Gansallo, Maria Bryant, Louise Padgett, Jessica Sheringham, Shahid Islam, Philip Garnett.

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
