## [Decision Letter · Decision Letter 0]

21 Oct 2025

Dear Dr. Bryant,

Thank you for submitting your manuscript to PLOS ONE. After careful consideration, we feel that it has merit but does not fully meet PLOS ONE’s publication criteria as it currently stands. Therefore, we invite you to submit a revised version of the manuscript that addresses the points raised during the review process.

We look forward to receiving your revised manuscript.

Kind regards,

Elizabeth McGill

Academic Editor

PLOS ONE

Journal Requirements:

2. As you did not seek ethical oversight for the current synthesis of existing reports, please could you remove all ethical approval documents from your submission.

“Authors MB, JS and SI received a UK Prevention Research Partnership

(MR/S037527/1) award for this study/project, which is funded by the British Heart Foundation, Cancer Research

UK, Chief Scientist Office of the Scottish Government Health and Social Care

Directorates, Engineering and Physical Sciences Research Council, Economic and

Social Research Council, Health and Social Care Research and Development Division

(Welsh Government), Medical Research Council, National Institute for Health

Research, Natural Environment

Research Council, Public Health Agency (Northern Ireland), The Health Foundation

and Wellcome.”

4. Thank you for stating the following in your manuscript:

“This work was supported by the UK Prevention Research Partnership (MR/S037527/1).”

“Authors MB, JS and SI received a UK Prevention Research Partnership

(MR/S037527/1) award for this study/project, which is funded by the British Heart Foundation, Cancer Research

UK, Chief Scientist Office of the Scottish Government Health and Social Care

Directorates, Engineering and Physical Sciences Research Council, Economic and

Social Research Council, Health and Social Care Research and Development Division

(Welsh Government), Medical Research Council, National Institute for Health

Research, Natural Environment

Research Council, Public Health Agency (Northern Ireland), The Health Foundation

and Wellcome.”

Additional Editor Comments (if provided):

Dear Authors

Thank you for your submission which will make a great contribution to the literature. Both reviewers found this paper well developed and written. They have made several suggestions to further strengthen it which I ask you take on board. In addition, please address the following:

- It would be helpful to have a little more detail on the methods: whose responsibility was it to update the ActEarly project Log and how? Given how many projects, were there much missing data and if so, how did the team approach this issue? Throughout this section, it is unclear who was doing what, how many researchers were involved in data collection and data analysis, etc. Please provide clarifying detail

- The cover letter links to the protocol, but this is not referenced in the text; please include

I look forward to receiving your revised manuscript.

Reviewers' comments:

Reviewer's Responses to Questions

**Comments to the Author**

1. Is the manuscript technically sound, and do the data support the conclusions?

Reviewer #1: Yes

Reviewer #2: Yes

2. Has the statistical analysis been performed appropriately and rigorously?

Reviewer #1: Yes

Reviewer #2: N/A

3. Have the authors made all data underlying the findings in their manuscript fully available?

Reviewer #1: Yes

Reviewer #2: Yes

4. Is the manuscript presented in an intelligible fashion and written in standard English?

Reviewer #1: Yes

Reviewer #2: Yes

Reviewer #1: PLOS ONE manuscript peer review:

“Mapping ActEarly: using a child health map to evaluate a City Collaboratory

programme on early promotion of good health and wellbeing”

Manuscript number: PONE-D-25-30109

This is a very important evaluation that took a systems mapping approach to understand where the ActEarly programme projects took place within the wider child health system. It found that ActEarly demonstrated good coverage across the wider system and mostly met the priority areas identified by stakeholders.

As the authors highlight, a key strength of the approach is that it has the potential to be replicated in other settings to support the evaluation of large-scale system-level public health programmes.

I can agree that, as per criteria for publication:

1. The study presents the results of original research.

2. The results reported have not been published elsewhere.

3. Experiments, statistics, and other analyses are performed to a high technical standard and are described in sufficient detail.

4. Conclusions are presented in an appropriate fashion and are supported by the data.

5. The article is presented in an intelligible fashion and is written in standard English.

6. The research meets all applicable standards for the ethics of experimentation and research integrity.

7. The article adheres to appropriate reporting guidelines and community standards for data availability.

In order for the paper to be published I make the following suggestions for the authors:

Abstract

Line 41/42 - suggest clarify to “provided activities across the full child health system and corresponding early life core outcome indicators that were prioritised by relevant individuals, organisations and community groups” . Or, if this has been misinterpreted, rephrase to make clear what you are saying here.

Line 44 – suggest a line to list examples of the ‘other information sources’ depending on word limit

Line 47/48 – if space then mention the 3 outcome indicators that projects did not map onto, i.e. priority areas for stakeholders that have been missed by ActEarly.

Introduction

The authors provide a coherent background to the study, outlining the history of the project clearly and the aims of the present study.

Suggest subheadings to introduce the CHM and COS .

Line 120- 125 – I would just question whether the ‘extent to which ActEarly reached its goal of enacting city-wide changes’ is the same as the extent to which ActEarly operates across different domains of a complex system of child health and wellbeing. Maybe for clarity here you could explain that the evaluation sought to explore the extent to which ActEarly met it’s aim of [x,y,z] through a systems evaluation which involved…

Line 125 – provide example(s) of where the CHM has been used before and how effective it is.

Methods

Line 175-176 - suggest clarifying the sentence, does this mean older adult and childhood poverty are the highest in the country? Or in comparison to the national average? Suggest adding a couple of stats to clarify.

Line 186 – I think this is supposed to be ‘Healthier Wealthier Families’

Line 212 – is Act Early the same age range as CHM? Suggest mentioning if it is/is not.

Table 1 – are social environment and physical environment criteria the wrong way round (e.g. would assume open space, parks, green space etc would be physical environment) and family and social relationships etc would be social environment). Suggest double checking.

Results

Line 234 – suggest listing initials of ‘research team’ for clarity.

Line 297 – delete ‘is’

Line 298 – suggest use consistent language with Table 4 (e.g. rephrase to “activities related to ‘public health’ factors”)

Figure 1 – It’s a potentially effective and useful visual, however I suggest making sure each project name can be clearly seen as currently some areas of high density cannot be read.

Discussion

Some discussion of the strengths and weaknesses of the CHM would be useful. Could be used in other contexts? How up-to-date is it? Are there domains that are missing or under or over-represented?

Reviewer #2: Thank you for the opportunity to review this paper. The approach taken by the authors is interesting and will make a great contribution to the literature. There are a few areas where the paper would benefit from more in-depth and critical engagement with its subject matter and the frameworks it has used. I list three cases that require particular attention below:

It would be good to see some more reflection on the systems map that was used, for example: for what purpose and in which context was this map developed? Who were the participants? And how might all three of these factors influence what is in- and excluded in the map? Then, you could reflect on how this links in with the current study. I.e. why was this map chosen? How does the purpose / context of the map link to that of ActEarly and are there some areas of focus for ActEarly that are not represented on the map?

I would question the assertion that connectivity in the map equals significance of that part of the system to the system as a whole, as seems suggested in line 243. The systems map used was inevitably a product of a specific time, context and set of participants, and so which nodes are most connected in this map do therefore not necessarily mean they are of most significance in childhood health.

Discussion would expect some more critical reflection. In particular, I would like to have seen reflection on:

- Are all projects that are reportedly on more social or economic aspects also more upstream in nature? I.e. do they not suffer from lifestyle drift where they may (for example) intend to target a households income status but do so by providing budget information.

- There are some factors in your table which are well-connected in the map but on which there are no ActEarly interventions. In the discussion you write that these are more likely to be focused on adults or behaviour change. But some of the ones in your table include social contact, smoking, positive friendship groups, etc. which

- You do briefly touch on that this method cannot show the impact that actions have had across the system. It would be good to expand on this either here or in the methods.

**Do you want your identity to be public for this peer review?** For information about this choice, including consent withdrawal, please see our Privacy Policy

Reviewer #1: **Yes**:  Anna Gibson

Reviewer #2: No

---

## [Author Response · Author response to Decision Letter 1]

5 Dec 2025

Please see a full response in the 'Response to Reviewers comments' document

---

## [Editor Report · Decision Letter 1]

18 Dec 2025

Mapping ActEarly: using a child health map to evaluate a City Collaboratory programme on early promotion of good health and wellbeing

PONE-D-25-30109R1

Dear Dr. Bryant,

We’re pleased to inform you that your manuscript has been judged scientifically suitable for publication and will be formally accepted for publication once it meets all outstanding technical requirements.

Kind regards,

Elizabeth McGill

Academic Editor

PLOS One

Additional Editor Comments (optional):

Dear Authors

Thank you for comprehensively taking on board the reviewer comments. The manuscript has been strengthened as a result and I look forward to seeing it published.
---

## [Editor Report · Acceptance letter]

PONE-D-25-30109R1

PLOS One

Dear Dr. Bryant,

I'm pleased to inform you that your manuscript has been deemed suitable for publication in PLOS One. Congratulations! Your manuscript is now being handed over to our production team.

Kind regards,

on behalf of

Dr Elizabeth McGill

Academic Editor

PLOS One